# Management of plant nutrient dynamics under alkaline soils through graded application of pressmud and gypsum

M. L. Dotaniya[1]*, M. D. Meena[1], R. L. Choudhary[1], M. K. Meena[1], Harvir Singh[1], C. K. Dotaniya[2], L. K. Meena[1], R. K. Doutaniya[3], K. N. Meena[1], R. S. Jat[1], P. K. Rai[1]

1 ICAR-Directorate of Rapeseed- Mustard Research, Bharatpur, India, 2 Department of Soil Science & Agricultural Chemistry, SKRAU, Bikaner, India, 3 Department of Agronomy, SKN College of Agriculture, Jobner, India

* mohan30682@gmail.com, Mohan.Dotaniya@icar.gov.in

**Data Availability Statement:** All relevant data are within the paper.

## Abstract

An incubation experiment was conducted to monitor the effect of different organic matter inputs with the graded application of gypsum at different time intervals on soil pH, sodium (Na) content and available plant nutrients like nitrogen (N) and sulphur (S) in alkaline soil. The experiment was formulated with nine treatments, *i.e.* control ($T_1$), recommended dose of fertilizer (RDF) ($T_2$), RDF+$Gyp_1$ ($T_3$), RDF+$FYM_5$+$Gyp_2$ ($T_4$), RDF+$FYM_{10}$+$Gyp_1$ ($T_5$), RDF+$PM_5$+$Gyp_2$ ($T_6$), RDF+$PM_{10}$+$Gyp_1$ ($T_7$), RDF+$FYM_{2.5}$+$PM_{2.5}$+$Gyp_2$ ($T_8$), RDF+$FYM_5$+$PM_5$+$Gyp_1$ ($T_9$) with three replications. Periodical soil samples were taken at six and twelve months intervals. Results showed that the addition of organic matter reduced the pH and Na content in the soil. More reduction was observed at one year period as compared to six months. The addition of farmyard manure (FYM) and pressmud (PM) at 10 t/ha with gypsum (1 t/ha) improved available N and available S content as compared to organic inputs (5 t/ha) with gypsum (2 t/ha) in soil. Pressmud application with FYM showed better availability of plant nutrients and a reduction of soil pH (8.39 to 7.79) and Na content from 626 to 391 *m*Eq/L in the soil during the incubation period. During the study, the application of treatment $T_9$ (FYM and PM in equal ratio with 1 t/ha gypsum) showed a better availability of available N (175 to 235 kg/ha) and S (15.44 to 23.24 kg/ha) and reduced the active ion concentration of Na. This study is very useful for the management of sodium toxicity, improving soil health and the mineralization rate of organic matter through the application of organic inputs for sustainable crop production.

## Introduction

Plant nutrients are critical to crop productivity, and nutrient application must be balanced for long-term agricultural production and soil health. The physico-chemical features of the soil and management parameters influence nutrient availability in the soil [1]. Higher concentrations of salt cations ions like sodium (Na), calcium (Ca), and magnesium (Mg) and associated chloride (Cl), sulphates ($SO_4$), carbonate ($CO_3$) and bicarbonate ($HCO_3$) anions limit the availability of important plant nutrients [2]. In India's dry and semi-arid regions, the problem is exacerbated

**Funding:** The authors received no specific funding for this work.

**Competing interests:** The authors have declared that no competing interests exist.

**Abbreviations:** CRD, Completely randomized design; $EC_e$, Extract electrical conductivity; FYM, Farm yard manure; Gyp, Gypsum; m ha, Million hectares; MSW, Municipal solid waste; mt, Million tonnes; N, Nitrogen; Na, Sodium; PM, Pressmud; RDF, Recommended dose of fertilizer; S, Sulphur.

significantly. Due to a lack of organic matter in the soil, the application of plant-available nutrients throughout the crop season does not produce the expected results. All soil microbial activity is based on organic matter, which improves the mineralization kinetics of given inputs. About 6.74 million hectares (m ha) of land in India are salt affected with a preponderance in particular places, such as the agriculturally significant Indo–Gangetic basin [3]. Among salt affected Indian soils, sodic area dominate on roughly 3.77 m ha, resulting in an annual loss of about 11.2 million tonnes (mt) of farm productivity valued at US$ 2.3 billion [4]. In saline/alkaline soils, salt ions impede the organic matter mineralization process and diminish the microbial population and diversity [5]. Under salt affected soils, crop plants have a limited nutrient supply capacity, according to researchers. It also said that environmental services were poor due to harsh climatic and soil conditions. Therefore, scientific management of salt affected soil, food production may be increased significantly and rationally without harming the environment or human health [6].

By 2050, India's population could exceed 1.6 billion people, requiring 333 mt of food grain production to feed the hungry. Fresh water availability per capita is declining, from 4000 $m^3$ in 1947 to 1683 $m^3$ in 2021 [7]. Farmers are obliged to utilize low-quality water for irrigation, resulting in soil contamination with heavy metals, organic pollutants, and salt [8]. This is one of the most significant obstacles to crop output and long-term sustainability [7]. Many examples can be found throughout history that indicate how the use of low-quality water exacerbated infertility and reduced crop productivity. The partial pressure of water is increased by salt ions, which limits water uptake from the soil to the plant. This process was also responsible for soil microorganisms' poor growth and survival. The addition of organic and inorganic components to saline and sodic soils is the need of the hour for long-term crop development [9].

The addition of organic matter to soil increased soil organic carbon (SOC), which is necessary for plant nutrient kinetics and soil biota growth [10]. Soil microbial organisms feed on soil carbon and are responsible for the rapid breakdown of organic materials due to an increased population. Different types of organic acids were released during the decomposition process. These features are mediated by soil chemical properties and improve soil aggregation [11], plant nutrients [12], soil conditions [13], and overall soil health. According to Dotaniya et al. [14], adding pressmud to sewage-irrigated soils reduced heavy metal labile fractions while increasing plant nutrient availability. Meena et al. [6] found that adding municipal solid waste (MSW) compost to soil increased the available S level and reduced salt ion toxicity in a similar way. Sugarcane manufacturers produce pressmud as a byproduct. Its potential usage in crop production systems is under-utilized in the agricultural segment [11]. It contains a lot of organic C as well as other plant nutrients. A trace amount of sugar increased the microbial population in the soil, speeding up the organic matter mineralization kinetics [15]. Farm yard manure (FYM) is an oldage agricultural feed that is used to boost SOC levels [16]. The majority of Indian soils have low to moderate levels of SOC, which is one of the key drivers for low plant nutrient utilization efficiency [17]. Higher levels of organic matter must be added to saline-sodic soils in order to boost yield. In saline soil, FYM increased the microbial population and crop output [6]. The supply of FYM is dwindling with the passage of time, necessitating the use of an alternative source of organic matter; and PM could be a reasonable alternative in this situation.

In this backdrop, a hypothesis was formulated to monitor the effect on organic matter addition (source and levels) on ionic chemistry and availability of major plant nutrient in alkaline soils.

## Materials and methods

### Experimental details

A laboratory experiment was conducted at ICAR-Directorate of Rapeseed-Mustard Research, Bharatpur, India to manage the plant nutrients under alkaline soil through pressmud and

gypsum application. Geo-referenced bulk soil samples were collected from nearby Kumher village of Bharatpur district, India. It is located 27.32°N and 77.37°E. In Bharatpur, the summers are scorching hot while the winters are pleasant. The temperature varies between 38°C and 45°C during the summer months of March to June. The temperature drops to roughly 27°C as the monsoon season (July–September) begins, with a humidity level of 70–75%. It is categorized under alkaline soil by the analysis of soil parameters. Soil pH was measured by saturation soil paste and extracted electrical conductivity as mentioned in Singh et al. [18]. The analyzed data showed pH (8.42), $EC_e$ (13.92 dS/m), 13.9 cmol($p^+$)kg$^{-1}$, $CO_3^{2-}$ + $HCO_3^-$ was (11.09 $m$Eq/L), $SO_4^{2-}$ (114.3 $m$Eq/L), $Ca^{2+}$ + $Mg^{2+}$ (41.23 $m$Eq/L) and $Na^+$ ions (632 $m$Eq/L) in soil. Soil plant nutrients like available N P K and S was measured as per the standard protocol mentioned in Singh et al. [18]. Most of the major plant nutrients were present in low to medium range of soil fertility. FYM was collected from the research farm of ICAR-DRMR, Bharatpur; and PM was collected from the Daurala Sugarmil, Meerut. The PM and FYM utilized in the research were also examined by standard analytical procedures and found that PM properties like pH (8.32) EC (3.21 dS/m), sodium content (0.014%), potassium concentration (0.037%), OC (9.55 mg/kg), total N (0.019%), total P (0.012 mg/kg) and sulphate-S (0.059%); whereas, FYM properties pH (7.24) EC (1.63), sodium content (ND), potassium concentration (0.042%), OC (14.09 mg/kg), total N (0.54%), total P (0.023 mg/kg) and S (0.012%). With the analysis data of soil samples and organic amendments; different treatment combinations were formulated by addition of FYM, PM and gypsum, *i.e.* control ($T_1$), RDF ($T_2$), RDF + $Gyp_1$ ($T_3$), RDF + $FYM_5$ + $Gyp_2$ ($T_4$), RDF + $FYM_{10}$ + $Gyp_1$ ($T_5$), RDF + $PM_5$ + $Gyp_2$ ($T_6$), RDF + $PM_{10}$ + $Gyp_1$ ($T_7$), RDF + $FYM_{2.5}$ + $PM_{2.5}$ + $Gyp_2$ ($T_8$), RDF + $FYM_5$ + $PM_5$ + $Gyp_1$ ($T_9$). The different abbreviation of treatments described as RDF (80:40:40 kg/ha NPK, respectively), $Gyp_1$ (gypsum application @ 1 t/ha), $Gyp_2$ (gypsum application @ 2 t/ha), $FYM_{10}$ (10 t/ha), $FYM_5$ (5 t/ha), $PM_{10}$ (10 t/ha), $PM_5$ (5 t/ha), $FYM_{2.5}$ (2.5 t/ha), $PM_{2.5}$ (2.5 t/ha). Bulk soil was filled into a plastic pot (500 g in each) and added different treatments as mentioned above. Soil moisture (at field capacity) and temperature (28 ± 1°C) was maintained at laboratory conditions. Addition of distilled water based on the water loss at a time interval. Periodical destructive soil samples were taken at six and one year time interval. After processing soil samples, different properties of soil like pH, sodium content, available plant nutrients like phosphorus in bulk soil [19]; whereas, available S [20] and N in incubated soils were measured by Subbiah and Asija [21] as mentioned in Singh et al. [18] to monitor the availability under different treatments.

## Statistical analysis

The different parameters were analysed and computed in various parameters data. These treatments were statistically analyzed in completely randomized design (CRD) with three replications as per method described in Gomez and Gomez [22]. The study of variance method (ANOVA) was used to assess the significance of treatment effects. At the $p \leq 0.05$ level of significance, the treatment means were evaluated using the least significant difference (LSD). For the treatment comparison Duncan's Multiple Range test (DMRT) was used in this experiment.

## Results and discussion

### Effect on soil pH

Application of FYM, pressmud and gypsum showed a significant effect on soil pH (Table 1). At six and 12 months time interval in control, RDF treated soils didn't show significantly difference on soil pH. However, addition of gypsum (1 t/ha) significantly ($p<0.05$) reduced the soil pH in both the time interval. However, further adding FYM levels 5 t/ha with gypsum (2 t/ha) and 10 FYM with gypsum (1 t/ha) reduced pH value 7.92 to 7.86, which was found non significant

**Table 1. Effect of FYM, pressmud and gypsum application on soil pH.**

| Treatment | Duration | | Mean |
|---|---|---|---|
| | **6 Months** | **12 Months** | |
| $T_1$ (Control) | 8.39$^a$ | 8.43$^a$ | 8.41 |
| $T_2$ (RDF) | 8.39$^a$ | 8.44$^a$ | 8.42 |
| $T_3$ (RDF+Gyp$_1$) | 8.03$^b$ | 7.94$^b$ | 7.99 |
| $T_4$ (RDF+FYM$_5$+Gyp$_2$) | 7.92$^{bc}$ | 7.92$^b$ | 7.92 |
| $T_5$ (RDF+FYM$_{10}$+Gyp$_1$) | 7.86$^{cd}$ | 7.85$^{bc}$ | 7.86 |
| $T_6$ (RDF+PM$_5$+Gyp$_2$) | 8.04$^b$ | 7.94$^b$ | 7.99 |
| $T_7$ (RDF+PM$_{10}$+Gyp$_1$) | 7.88$^{cd}$ | 7.80$^{bcd}$ | 7.84 |
| $T_8$ (RDF+FYM$_{2.5}$+PM$_{2.5}$+Gyp$_2$) | 7.82$^{cd}$ | 7.77$^{cd}$ | 7.80 |
| $T_9$ (RDF+FYM$_5$+PM$_5$+Gyp$_1$) | 7.79$^d$ | 7.70$^d$ | 7.75 |

*Treatment means in a column with the letters in common are not significant by Duncan's New Multiple Range Test at 5% level of significance. RDF (80:40:40::NPK), Gyp$_1$ (gypsum application @ 1 t/ha), Gyp$_2$ (gypsum application @ 2 t/ha), FYM$_{10}$ (10 t/ha), FYM$_5$ (5 t/ha), PM$_{10}$ (10 t/ha), PM$_5$ (5 t/ha), FYM$_{2.5}$ (2.5 t/ha), PM$_{2.5}$ (2.5 t/ha)

difference (p < 0.05). A similar pattern, the addition of pressmud at 5 t/ha along with RDF and gypsum at 2 t/ ha reduced the soil pH. Further increased in PM application rate (PM increased 5 to 10 t/ha) and gypsum (gypsum 1 t/ha) pH value decreased to 7.88 from 8.42. The addition of organic matter through the FYM at 5 t/ha and PM at 5 t/ha with one t /ha gypsum reduced the maximum soil pH. Among the comparison of all the treatments, the application of the higher amount of organic matter act as a powerful soil pH reducer. FYM and PM both are equally performing with respect to soil pH value. One year after the incubation period soil samples were analyzed and showed that the addition of a higher amount of FYM from 5 t/ha to 10 t/ha reduced the soil pH from 7.92 to 7.85; whereas, PM application rate from 5 to 10 t/ha with graded application of gypsum reduced more pH units (7.94 to 7.80) at 12 months incubation period. Maximum pH was reduced under treatment $T_9$ (RDF + FYM$_5$ + PM$_5$ + Gyp$_1$) and *at par* with the treatment $T_8$ (RDF + FYM$_{2.5}$ + PM$_{2.5}$ + Gyp$_2$) during the experiment. The mean value of the experiment showed that higher pH was under RDF and increasing the gypsum and organic matter graded application reduced the soil pH. The addition of organic matter along with gypsum in soil enhances the microbial population and produce different types of organic acids which mobile the plant nutrient by reducing the soil pH. The addition of FYM at 10 t/ha reduced the soil pH and enhanced the crop yield. Different types of soil micro-organisms is reduced salt toxicity and improve plant nutrient availability to plants [23]. This sort of increased EC is possible, because the decomposition processes of organic matter favor the accumulation of carbon di oxide ($CO_2$) and release of large amount of salts in solution which results in higher EC [24]. Kalaivanan and Hattab [12] reported that pressmud application at 5 t/ha didn't affect the soil pH and electrical conductivity (EC) did not show any marked variation with the application of enriched pressmud compost. Prabhavathi and Parama [25] reported that the addition of pressmud at 10 t/ha along with RDF reduced the soil pH under finger millet crop. Sheoran et al. [9] reported that pressmud application (10 Mg/ha) led to appreciable reductions in soil pH (1.6–3.6%) and exchangeable sodium percentage (ESP; 10.4–20.1%) with concomitant improvements in plant physiological and yield-related traits across different alkali soil.

## Effect on soil sodium content

The addition of FYM and pressmud in a combination of graded dose of gypsum was very much affected by the duration and soil amendments. Increasing the application rate of FYM

**Table 2. Effect of FYM, pressmud and gypsum application on sodium content ($m$Eq/L).**

| Treatment | Duration | | Mean |
|---|---|---|---|
| | **6 Months** | **12 Months** | |
| $T_1$ (Control) | 626[a] | 627[a] | 626 |
| $T_2$ (RDF) | 629[a] | 633[a] | 631 |
| $T_3$ (RDF+Gyp$_1$) | 618[a] | 590[b] | 604 |
| $T_4$ (RDF+FYM$_5$+Gyp$_2$) | 598[a] | 549[c] | 573 |
| $T_5$ (RDF+FYM$_{10}$+Gyp$_1$) | 544[b] | 529[d] | 537 |
| $T_6$ (RDF+PM$_5$+Gyp$_2$) | 543[b] | 509[e] | 526 |
| $T_7$ (RDF+PM$_{10}$+Gyp$_1$) | 514[b] | 484[f] | 499 |
| $T_8$ (RDF+FYM$_{2.5}$+PM$_{2.5}$+Gyp$_2$) | 507[b] | 442[g] | 475 |
| $T_9$ (RDF+FYM$_5$+PM$_5$+Gyp$_1$) | 398[c] | 384[h] | 391 |

*Treatment means in a column with the letters in common are not significant by Duncan's New Multiple Range Test at 5% level of significance. RDF (80:40:40::NPK), Gyp$_1$ (gypsum application @ 1 t/ha), Gyp$_2$ (gypsum application @ 2 t/ha), FYM$_{10}$ (10 t/ha), FYM$_5$ (5 t/ha), PM$_{10}$ (10 t/ha), PM$_5$ (5 t/ha), FYM$_{2.5}$ (2.5 t/ha), PM$_{2.5}$ (2.5 t/ha)

upto 5 t/ha along with 2 t/ha gypsum reduced the sodium content (598 $m$Eq/L) in soil but it is not significant (p<0.05) at six month time interval over control (Table 2). Increasing the PM upto 10 t/ha with gypsum (1 t/ha) reduced significant (p<0.05) amount of sodium (514 $m$Eq/L) over control (626 $m$Eq/L), $T_2$ (629 $m$Eq/L), $T_3$ (598 $m$Eq/L) and $T_4$ (544 $m$Eq/L) at 6 month time interval. The highest effective sodium reduction was observed in treatment $T_9$ (RDF+FYM$_5$+PM$_5$+Gyp$_1$) which are 13.1%, 17.9% and 36.4% in $T_5$, $T_7$ and $T_9$, respectively. Similar pattern application of soil amendments in this experiment and after analysis of sodium content after one year incubation period and found that application of gypsum in the addition to RDF mediated the sodium content from 627 to 633 mEq/L after 12 months incubation period. Further, the addition of FYM levels (5 and 10 t/ha), PM (5 to 10 t/ha) with gypsum levels 1 and 2 t/ha; PM reduced more sodium ions than a similar amount of FYM application with gypsum at 12 months duration. In another treatments, like addition of organic amendment at 5 t/ha through FYM, PM and a half of the dose through PM and half from FYM with gypsum at 2 t/ha reduced the sodium content. However, apart from $T_9$ maximum Na ions were reduced in treatment $T_8 > T_6 > T_4$ were 29.5%, 18.8% and 12.4%, respectively. The mean of both the time period on sodium content with respect to the treatment was also showed lower down the sodium with the addition of organic and gypsum substances. The treatment $T_9$, which had half of the organic matter added by FYM and the rest by PM with gypsum (1 t/ha) application during the experiment, had the greatest reduction in sodium content.

Pressmud and molasses were utilized as soil additives in the early twentieth century. The importance of pressmud as an organic manure for use in agriculture has long been known, as it contains vital plant nutrients in an organic form while also acting as a powerful soil ameliorant [26]. Alkaline soils are characterized by clay complexes with high proportions of sodium ($Na^+$) on their exchange and alkaline soil reaction [27] that together have detrimental effects on soil aeration and water movement [28], root development [29], plant seedling emergence and enzymatic activities [30], resulting in poor crop yields [31, 32]. Application of gypsum with the organic soil amendments dispersed the $Na^+$ ions and flocculating the Ca on exchange sites and improves the soil health [31]. It also observed that during the decomposition of organic substances produced low molecular organic acids, which lead the soil aggregates formation, better soil structure and leaching of Na ions by water transportation [11]. It was also

reported that addition of 10 Mg/ ha significantly reduced the exchangeable sodium percentage in rice–wheat cropping system [33].

## Effect on available N

Nitrogen mineralization kinetics in the soil is very much affected by the addition of organic matter and gypsum. During the experiment, it was observed that nitrogen concentration was almost equal to the $T_1$ (180 kg/ha) and $T_2$ (191 kg/ha) treatment as measured in $T_3$ (190 kg/ha); whereas, the addition of gypsum application with organic soil amendments through FYM and PM enhanced the available N concentration in soil (Table 3). Application of FYM at 10 t/ ha improved the available N content from 203 to 218 kg/ha by the addition of gypsum 1t /ha. Comparison month the organic treatment at the 10 t /ha application with 1 t/ha gypsum application, both the organic amendment performed at par and showed value 218 in $T_5$ and 217 in $T_7$. The highest available N (230 kg/ha) was reported by the addition of organic matter FYM at 5 t/ha and 5 t/ha PM along the 1 t/ha gypsum application in $T_9$ treatment.

At 12 months of the incubation period, mineralization rate of applied the SOC enhanced by the source and graded application of gypsum. Addition of 5 t/ha FYM with 1 t/ha gypsum enhanced available N 8.8% over control. Among the delivery of organic matter through different sources at 5 t/ha with 2 t/ha gypsum application significantly affected the available N content in soil. Addition of organic matter at 5 t/ha through FYM, PM and 50: 50 ratio of FYM and PM reported higher available N 20%, 15.3%, 28.2% over to control, respectively. Similar pattern application of 10 t/ha of organic matter through FYM, PM and mixing of an equal amount of PM and FYM along with 1 t/ha gypsum application enhanced the available N 34%, 34.1% and 41.2% higher available N in soil over control, respectively. Here, result showed that FYM and PM both were significantly (0.05%) at par in available N over control. Conjugation of equal amounts (50:50) of FYM and PM enhanced the available N as a single application of organic matter through FYM and PM (each 10 t/ha). The mean available N during the experiment was much affected by the organic sources and graded level of gypsum. Highest mean available N was measured in treatment $T_9$ (RDF + $FYM_5$ + $PM_5$ + $Gyp_1$) 235 kg/ha as compared to control (175 kg/ha), RDF (190 kg/ha), in $T_5$ (223 kg/ha) and $T_7$ (223 kg/ha).

The addition of organic matter through PM application improves the soil microbial population and diversity enhancing the mineralization kinetics of SOM and mediated available N

**Table 3. Effect of FYM, pressmud and gypsum application on soil available N (kg/ha).**

| Treatment | Duration | | Mean |
|---|---|---|---|
| | **6 Months** | **12 Months** | |
| $T_1$ (Control) | 180[e] | 170[g] | 175 |
| $T_2$ (RDF) | 191[de] | 189[ef] | 190 |
| $T_3$ (RDF+$Gyp_1$) | 190[e] | 185[f] | 188 |
| $T_4$ (RDF+$FYM_5$+$Gyp_2$) | 203[cd] | 204[d] | 204 |
| $T_5$ (RDF+$FYM_{10}$+$Gyp_1$) | 218[ab] | 228[b] | 223 |
| $T_6$ (RDF+$PM_5$+$Gyp_2$) | 205[bc] | 196[e] | 201 |
| $T_7$ (RDF+$PM_{10}$+$Gyp_1$) | 217[b] | 229[b] | 223 |
| $T_8$ (RDF+$FYM_{2.5}$+$PM_{2.5}$+$Gyp_2$) | 213[bc] | 218[c] | 216 |
| $T_9$ (RDF+$FYM_5$+$PM_5$+$Gyp_1$) | 230[a] | 240[a] | 235 |

*Treatment means in a column with the letters in common are not significant by Duncan's New Multiple Range Test at 5% level of significance. RDF (80:40:40::NPK), $Gyp_1$ (gypsum application @ 1 t/ha), $Gyp_2$ (gypsum application @ 2 t/ha), $FYM_{10}$ (10 t/ha), $FYM_5$ (5 t/ha), $PM_{10}$ (10 t/ha), $PM_5$ (5 t/ha), $FYM_{2.5}$ (2.5 t/ha), $PM_{2.5}$ (2.5 t/ha)

dynamics in soil [34]. Conversion of raw pressmud into enriched pressmud compost through the composting process was increased the available nutrient content like N, P and K. Application of pressmud compost was increased the crop yield and soil properties than raw pressmud [35]. The organic matter added through PM application stimulation of microbial activity could have increased the partial pressure of carbon dioxide ($pCO_2$) in the soil solution [36], lowering soil pH and improving the calcium levels, which enhance the plant nutrient dynamics. Sultana et al. [37] reported that incorporation of PM in the combination of 50 percent MSW compost improved available nitrogen content 2–3 times higher in aerobic conditions. Further, described that increasing the mineralization period improved the concentration and availability of N ($NO_3^- $-N and $NH_4^+$-N) in soil.

## Effect on available S content

Available S content in soil was mediated by the application of organic matter in the soil. At six month time interval, the addition of 5 t/ha FYM with 2 t/ha gypsum at par response with RDF and RDF + gypsum at 1 t/ha, whereas, significant improvement over control treatment (Table 4). Treatment $T_5$ (RDF + $FYM_{10}$ + $Gyp_1$) showed significant improvement in available S (25.1%) over control, this improvement was higher (33.5%) with the application of PM along with 1 t/ha gypsum. Further supplied the 10 t/ha organic matter through the half by the FYM and remaining by the PM with 1 t/ha gypsum ($T_9$) showed at par response as $T_7$. A similar pattern, the available S mineralization rate was calculated at one year interval and found that increasing the organic matter dose, type and duration significantly affected the S availability in soil. Treatment $T_3$ improved the S content from 15.02 to 18.05 kg/ha at 12 months incubation period.

Whereas, application of 5 t/ha organic matter through FYM ($T_4$) and PM ($T_6$) with 2 t/ha gypsum showed at par results. Split application of FYM and PM each 2.5 t/ha with 2 t/ha gypsum ($T_8$) showed better availability of S in soil over alone application of FYM ($T_4$) and PM ($T_6$). The addition of 10 t/ha organic matter through FYM, PM and with the split application (FYM and PM) showed significant improvement in available S 20.65, 21.71 and 24.70 kg/ha at 12 months time interval. This showed that PM application improves the S content in the soil. The mean value of the experiment showed that the addition of organic matter improved the sulphur was 17.60, 18.09, 18.99, 20.23, 19.15, 21.44, 21.03 and 23.24 kg/ha in $T_2$, $T_3$, $T_4$, $T_5$, $T_6$,

**Table 4. Effect of FYM, pressmud and gypsum application on soil available S (kg/ha).**

| Treatment | Duration | | Mean |
|---|---|---|---|
| | **6 Months** | **12 Months** | |
| $T_1$ (Control) | 15.85[e] | 15.02[g] | 15.44 |
| $T_2$ (RDF) | 17.94[d] | 17.26[f] | 17.60 |
| $T_3$ (RDF+$Gyp_1$) | 18.12[d] | 18.05[e] | 18.09 |
| $T_4$ (RDF+$FYM_5$+$Gyp_2$) | 18.83[cd] | 19.14[d] | 18.99 |
| $T_5$ (RDF+$FYM_{10}$+$Gyp_1$) | 19.81[bc] | 20.65[c] | 20.23 |
| $T_6$ (RDF+$PM_5$+$Gyp_2$) | 19.08[cd] | 19.22[d] | 19.15 |
| $T_7$ (RDF+$PM_{10}$+$Gyp_1$) | 21.16[a] | 21.71[b] | 21.44 |
| $T_8$ (RDF+$FYM_{2.5}$+$PM_{2.5}$+$Gyp_2$) | 20.98[ab] | 21.07[bc] | 21.03 |
| $T_9$ (RDF+$FYM_5$+$PM_5$+$Gyp_1$) | 21.77[a] | 24.70[a] | 23.24 |

*Treatment means in a column with the letters in common are not significant by Duncan's New Multiple Range Test at 5% level of significance. RDF (80:40:40::NPK), $Gyp_1$ (gypsum application @ 1 t/ha), $Gyp_2$ (gypsum application @ 2 t/ha), $FYM_{10}$ (10 t/ha), $FYM_5$ (5 t/ha), $PM_{10}$ (10 t/ha), $PM_5$ (5 t/ha), $FYM_{2.5}$ (2.5 t/ha), $PM_{2.5}$ (2.5 t/ha)

$T_7$, $T_8$ and $T_9$ over control (15.44 kg/ha), respectively. Pressmud and gypsum contain a significant amount of S which improves the soil solution S concentration.

Application of S fertilizer in salt affected soils is a viable procedure to counteract the uptake of unnecessary toxic elements ($Na^+$ and $Cl^-$), which encourage selectivity of K or Na and the ability of calcium ions to decrease the harmful impacts of sodium ions in plants [38, 39]. The availability of S content in soil significantly increased because PM is also a source of S in soil during experiment; PM and FYM having slow release of plant nutrients resulted in higher residual S availability [15]. Further, microbial decomposition could release essential nutrients for plant uptake when the readily available organic C is from the PM [40]. Sultana et al. [37] predicted the plant nutrient dynamics through a kinetic model and indicated that incorporation of poultry manure and sugar pressmud had a higher capability to supply S to the crops. The combined use of biological (beneficial microbes) and organic materials (e.g., compost and straw) as bio-organic amendments with gypsum has great potential in ameliorating saline/alkaline soils [41]. Sulfuric acid is created in calcareous soils by the action of a genus of autotrophic bacteria, which reacts with native calcium carbonate to make soluble calcium and lower sodium toxicity [42]. Addition of OM through PM improved the microbial community in the soil and acted as a Na ion mediator [34]. However, it also increases the amount of S in the soil solution [41]. Addition of green manuring along with 2.5 t/ha gypsum reduced the soil pH and improved the plant nutrient availability (N, P, K, S and micronutrients) and biological properties (DHA, SMBC) over a period [43, 44]. In similar line, Ahmed et al. [45] reported that addition of pressmud and elemental S improved the soil properties and enhanced productivity of wheat and pearl millet cropping system. It was also observed that addition of 5% FYM and 5% PM in 12 dS/m improved the plant nutrient dynamics and enhanced the protein and micronutrient concentration in rice crop [46].

## Conclusions

Alkaline soil has a higher concentration of salt ions, which limits nutrient availability during crop growth. The addition of organic and inorganic substances to soil increases the rate of SOM mineralization and the concentration of plant nutrients in soil solution. In the present experiment, graded level of FYM, PM, and gypsum were applied to alkaline soil; and monitored the interaction effect on soil pH, Na content, and accessible plant nutrients such as N and S was computed. The results demonstrated that a six-month time period was associated with a reduced rate of organic matter mineralization and lower plant nutrient availability. After one year incubation period PM showed at par or better availability of available N and S and lowered the soil pH and sodium content in soil. Application of RDF along with 10 t/ha organic sources (equally contributed by FYM and PM) and 1 t/ha gypsum as a chemical soil amendment ($T_9$) showed better results with respect to lower the sodium content and enhancing the plant nutrient content (N and S) in soil. The findings of this research will be helpful for managing the alkaline soils by the application of PM with graded level of gypsum for sustainable crop production.

## Acknowledgments

Authors are thankful to technical and supporting staff of Natural Resource Management Unit, ICAR-Directorate of Rapeseed-Mustard Research, Bharatpur for valuable help during the course of study.

## Author Contributions

**Conceptualization:** M. L. Dotaniya, M. D. Meena, R. L. Choudhary, R. S. Jat, P. K. Rai.

**Data curation:** M. L. Dotaniya, M. K. Meena, C. K. Dotaniya.

**Formal analysis:** M. L. Dotaniya, M. D. Meena, C. K. Dotaniya, R. K. Doutaniya.

**Funding acquisition:** R. S. Jat, P. K. Rai.

**Investigation:** M. L. Dotaniya, M. D. Meena, M. K. Meena, K. N. Meena.

**Methodology:** M. L. Dotaniya, M. D. Meena, R. L. Choudhary, M. K. Meena, Harvir Singh, C. K. Dotaniya.

**Project administration:** M. L. Dotaniya, L. K. Meena, R. S. Jat, P. K. Rai.

**Resources:** Harvir Singh, L. K. Meena.

**Software:** C. K. Dotaniya, L. K. Meena, R. K. Doutaniya.

**Supervision:** M. L. Dotaniya, M. D. Meena, R. K. Doutaniya, K. N. Meena, R. S. Jat, P. K. Rai.

**Validation:** M. L. Dotaniya, M. D. Meena, P. K. Rai.

**Visualization:** R. K. Doutaniya.

**Writing – original draft:** M. L. Dotaniya, C. K. Dotaniya, R. K. Doutaniya.

**Writing – review & editing:** M. L. Dotaniya, M. D. Meena.

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
