## [Decision Letter · Decision Letter 0]

7 Jun 2023

PONE-D-23-16195Management of Plant Nutrient dynamics under Alkaline Soils through graded application of  Pressmud and GypsumPLOS ONE

Dear Dr. Dotaniya,

Thank you for submitting your manuscript to PLOS ONE. After careful consideration, we feel that it has merit but does not fully meet PLOS ONE’s publication criteria as it currently stands. Therefore, we invite you to submit a revised version of the manuscript that addresses the points raised during the review process.

We look forward to receiving your revised manuscript.

Kind regards,

Abhay Omprakash Shirale, PhD

Academic Editor

PLOS ONE

Journal Requirements:

   "The author(s) received no specific funding for this work. We will try to manage in due course of time."

Reviewers' comments:

Reviewer's Responses to Questions

**Comments to the Author**

1. Is the manuscript technically sound, and do the data support the conclusions?

Reviewer #1: Yes

Reviewer #2: Yes

2. Has the statistical analysis been performed appropriately and rigorously? 

Reviewer #1: Yes

Reviewer #2: Yes

3. Have the authors made all data underlying the findings in their manuscript fully available?

Reviewer #1: Yes

Reviewer #2: No

4. Is the manuscript presented in an intelligible fashion and written in standard English?

Reviewer #1: Yes

Reviewer #2: Yes

5. Review Comments to the Author

Reviewer #1: Dear Author's,

I have gone through manuscript (MS) entitled “Management of Plant Nutrient dynamics under Alkaline Soils through graded application of Pressmud and Gypsum,"" and made the necessary corrections in the MS text. The study is very useful to use of organic amendments particularly in the alkaline soil. Please go through the corrections as included in MS.

Reviewer #2: I read “Management of Plant Nutrient dynamics under Alkaline Soils through graded application of Pressmud and Gypsum” with a great interest. It is scientifically written as per the journal guidelines. I recommend minor revision based on the following points

1. Is it a field or laboratory experiment?….indicate in material and method part.

2. In results and discussion section, there are different ways of representing the units for the one. Please, unify the written units. For example, ton/ha and t/ha

3. Check unit of sodium ions

4. Table 1: You have superscripts (a,b, c). What do they refer?

5. Fig. 1, Add the error bars.

6. Fill texture in fig 1 & 2

7. Carefully check reference in text and reference section.

8. Comple/clearly the reference 4 in text.

9. Please, update the introduction and discussion with the recent literature.

https://doi.org/10.1016/B978-0-12-821556-2.00001-3

https://doi.org/10.1016/B978-0-323-90051-5.00019-

https://doi.org/10.1007/978-3-030-97000-0_18

10.Conclusion section should be updated and more clear with significant findings.

6. PLOS authors have the option to publish the peer review history of their article (what does this mean?). If published, this will include your full peer review and any attached files.

Reviewer #1: No

Reviewer #2: **Yes: **Dr. Alaa El Din Mahmoud

---

## [Author Response · Author response to Decision Letter 0]

8 Jun 2023

MS revised as per the directions of EIC and reviewers. A letter to EIC also attached with this submission.

---

## [Decision Letter · Decision Letter 1]

4 Jul 2023

Management of Plant Nutrient dynamics under Alkaline Soils through graded application of Pressmud and Gypsum

PONE-D-23-16195R1

Dear Dr. Dotaniya,

We’re pleased to inform you that your manuscript has been judged scientifically suitable for publication and will be formally accepted for publication once it meets all outstanding technical requirements.

Kind regards,

Abhay Omprakash Shirale, PhD

Academic Editor

PLOS ONE

Reviewers' comments:

Reviewer's Responses to Questions

**Comments to the Author**

1. If the authors have adequately addressed your comments raised in a previous round of review and you feel that this manuscript is now acceptable for publication, you may indicate that here to bypass the “Comments to the Author” section, enter your conflict of interest statement in the “Confidential to Editor” section, and submit your "Accept" recommendation.

Reviewer #1: All comments have been addressed

Reviewer #2: All comments have been addressed

2. Is the manuscript technically sound, and do the data support the conclusions?

Reviewer #1: Yes

Reviewer #2: Yes

3. Has the statistical analysis been performed appropriately and rigorously? 

Reviewer #1: Yes

Reviewer #2: Yes

4. Have the authors made all data underlying the findings in their manuscript fully available?

Reviewer #1: Yes

Reviewer #2: No

5. Is the manuscript presented in an intelligible fashion and written in standard English?

Reviewer #1: Yes

Reviewer #2: Yes

6. Review Comments to the Author

Reviewer #1: I have gone through the MS, the author has incorporated all the suggestions and improved. Now MS seems good.

Reviewer #2: The authors have answered all inquiries. The manuscript is well written and further explanations have been added. It can be accepted.

7. PLOS authors have the option to publish the peer review history of their article (what does this mean?). If published, this will include your full peer review and any attached files.

Reviewer #1: No

Reviewer #2: **Yes: **Dr. Alaa El Din Mahmoud

---

## [Editor Report · Acceptance letter]

28 Jul 2023

PONE-D-23-16195R1 

Management of Plant Nutrient dynamics under Alkaline Soils through graded application of  Pressmud and Gypsum 

Dear Dr. Dotaniya:

I'm pleased to inform you that your manuscript has been deemed suitable for publication in PLOS ONE. Congratulations! Your manuscript is now with our production department. 

Kind regards, 

on behalf of

Dr. Abhay Omprakash Shirale 

Academic Editor

PLOS ONE